# Relative cluster entropy for power-law correlated sequences

Anna Carbone[1] and Linda Ponta[2]

**1** Politecnico di Torino, Italy
**2** LIUC Castellanza, Italy

## Abstract

We propose an information-theoretical measure, the *relative cluster entropy* $\mathcal{D}_C[P\|Q]$, to discriminate among cluster partitions characterised by probability distribution functions $P$ and $Q$. The measure is illustrated with the clusters generated by pairs of fractional Brownian motions with Hurst exponents $H_1$ and $H_2$ respectively. For subdiffusive, normal and superdiffusive sequences, the relative entropy sensibly depends on the difference between $H_1$ and $H_2$. By using the *minimum relative entropy* principle, cluster sequences characterized by different correlation degrees are distinguished and the optimal Hurst exponent is selected. As a case study, real-world cluster partitions of market price series are compared to those obtained from fully uncorrelated sequences (simple Browniam motions) assumed as a model. The *minimum relative cluster entropy* yields optimal Hurst exponents $H_1 = 0.55$, $H_1 = 0.57$, and $H_1 = 0.63$ respectively for the prices of DJIA, S&P500, NASDAQ: a clear indication of non-markovianity. Finally, we derive the analytical expression of the relative cluster entropy and the outcomes are discussed for arbitrary pairs of power-laws probability distribution functions of continuous random variables.

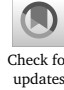

## 1 Introduction

Flow of information in complex systems with interacting components can be quantified via entropy measures [1–7]. In this context, discriminating between empirical data and models in

terms of information content is interesting from several viewpoints. Consider an experiment with the outcomes obeying the probability distribution $P$ whereas the distribution $Q$ is a model for the same experiment. Quantifying the error of the wrong assumption of the model compared to the empirical information content is relevant to a broad class of phenomena [8,9]. Such information-theoretical concepts bring also together the thermodynamic implications intrisically related to the evolution of the system under investigation. The dynamic of the information transferred along subsequent transformative states of a complex system can be described in terms of divergence of the probability distributions $P$ at time $t$ and $P'$ at a subsequent time $t'$. Hence, information-theoretical tools find applications in fields as diverse as climate, turbulence, neurology, biology and economics [10–13] and are increasingly adopted in unsupervised learning of unlabelled data where similarity/dissimilarity measures are concerned with dynamic rather than static features of the clustered data [14–16].

A recently proposed information measure, with the ability to quantify heterogeneity and dynamics of long-range correlated processes in a broad range of application areas, is the *cluster entropy* $\mathcal{S}_C[P]$ [17–20]. The measure has been defined as a Shannon functional with $P$ the power-law probability distribution of the clusters formed in a long-range correlated data sets. If $P$ is a distribution concentrated on a single cluster value, $\mathcal{S}_C[P] = 0$ corresponds to the minimum uncertainty on the outcome of the cluster size, the random variable of interest. If $P$ is a fully developed power-law distribution, $\mathcal{S}_C[P] = \ln N$ corresponds to the maximum uncertainty obtained as the power-law distribution spreads over a broad range of cluster values. Thus, according to the Shannon interpretation, $\mathcal{S}_C[P]$ can be understood as a measure of uncertainty of all the possible cluster outcomes. By extending the definition to continuous variables, the *differential cluster entropy* $S_C[P]$ added clues to the approach by clarifying the interplay of the different terms entering the cluster entropy and thus the origin of the excess randomness.

In this work, we go beyond the simple *measure of uncertainty* of the random variable outcomes provided by $\mathcal{S}_C[P]$. An inference method for hypothesis testing of a general class of models underlying a relevant stochastic process is developed. We put forward the *relative cluster entropy* or *cluster divergence* $\mathcal{D}_C[P\|Q]$ with the first argument $P$ the empirical distribution and the second argument $Q$ a model within a broad class of probability distributions. $\mathcal{D}_C[P\|Q]$ is a metric on the space of probability distributions, interpreted as a divergence rather than as a distance since it does not obey symmetry and triangle inequality. The asymmetry of the relative entropy reflects the asymmetry between data and models, hence it can be used for inference purposes on the model underlying a given distribution. If $\mathcal{D}_C[P\|Q] >> 0$, the hypothesis likelihood is very low and, unless the quality of the empirical data should be questioned, the model distribution $Q$ must be rejected. The higher $\mathcal{D}_C[P\|Q]$, the lower the likelihood of the hypothesis. If the hypothesis on the model were true, $P$ should fluctuate around its expected value $Q$, with fluctuations of limited amplitude and occurrence probability greater than the significance level, resulting in the acceptance of the model $Q$.

The *cluster entropy* $\mathcal{S}_C[P]$ and the *relative cluster entropy* $\mathcal{D}_C[P\|Q]$ can be interpreted as information measures over partitions generated by a coarse-grained mapping of the two-dimensional phase-space spanned by a particle, e.g. a simple Brownian path described by the random variable $\{x_t\}$. According to Gibbs' original idea at the core of the information entropy concept, a coarse grained description is defined by smoothing out fine details and increasing the observer's ignorance about the exact microstate of the system. As the structure description becomes blurrier, randomness and entropy increase. A coarse-grained description is obtained by performing a local average over the phase-space cells with increasing size. In the information clustering approach adopted here, the coarse grained description of the particle path $\{x_t\}$ is obtained by a local average $\{\widetilde{x}_{t,n}\}$ over the phase-space cells with the parameter $n$ defining the cell sizes. The regression $x_t = \widetilde{x}_{t,n} + \epsilon_{t,n}$ yields the errors $\epsilon_{t,n} = x_t - \widetilde{x}_{t,n}$ which

ultimately generate a finite partition $\{\mathcal{C}\} = \{\mathcal{C}_{n,1}, \mathcal{C}_{n,2}, \ldots, \mathcal{C}_{n,j}\}$ for each $n$. The partition process generates regions, named as *clusters*, bounded between the values of $t$ when $\epsilon_{t,n} = 0$, which correspond to complete information with minimum entropy. The probability distribution functions of the random variables defined by $\epsilon_{t,n}$ univocally quantify the loss of structure and information of the coarse grained representation. As already noted, the *cluster entropy* $\mathcal{S}_\mathcal{C}[P]$ is bounded, involves integrating over cell components, ranges from the minimum to the maximum value as the description ranges from the finest-grained (corresponding to the smallest clusters) to the coarsest-grained partition (corresponding to the largest clusters). The *relative cluster entropy* $\mathcal{D}_\mathcal{C}[P\|Q]$ is also bounded, involves integration over cells, ranges between a maximum value, depending on the two distributions, and the minimum value 0 for $P = Q$.

The ability of the cluster divergence $\mathcal{D}_\mathcal{C}[P\|Q]$ to select an optimal distribution could be relevant in several contexts. In particular, complex phenomena obeying power-law distributions are still raising concerns regarding accuracy and veracity of the estimation of the power law exponent [21]. To illustrate how the *relative cluster entropy* operates, synthetic and real-world data featuring power-law distribution behavior are considered. First, the approach is implemented on pairs of synthetic fractional Brownian motions (*fBms*) with given Hurst exponent. A systematic dependence of $\mathcal{D}_\mathcal{C}[P\|Q]$ on the Hurst exponents of the pair is found. The *minimum relative entropy* principle is then implemented as a selection criterion to extract the optimal correlation exponent of the sequence. Second, as a real-world case, we study the divergence $\mathcal{D}_\mathcal{C}[P\|Q]$ of financial price series. The probability distribution $P$ is obtained by ranking the clusters generated in each price time series and compared to the distribution $Q$ drawn from synthetic *fBms* data adopted as model. The *minimum relative entropy* principle yields the best estimate of the correlation exponents of the financial series and quantifies the deviation of the price series from the assumed model.

The manuscript is organized as follows. In Section 2 the main computational steps of the *relative cluster entropy* method are described for discrete variables. The approach is illustrated for synthetic (fractional Brownian motions) and real-world (market price series) data. In Section 3 the *relative cluster entropy* is extended to continuous random variables, conclusions and suggestions for further developments are drawn.

## 2 Methods and Results

In this section, the main steps of the *relative cluster entropy* approach are described. The interest is towards the development of a divergence measure able to evaluate the situation where a model probability distribution $Q$ is defined in parallel to the true probability distribution function $P$ of the cluster partition. Before illustrating how the proposed *cluster divergence* works, a few definitions are recalled.

Consider the time series $\{x_t\}$ of length $N$ and the local average $\widetilde{x}_{t,n} = \frac{1}{n}\sum_{n'=0}^{n-1} x(t-n')$ of length $N-n$ with $n \in (1,N)$. For each $n$, a partition $\{\mathcal{C}\}$ of non-overlapping clusters is generated between consecutive intersections of $\{x_t\}$ and $\{\widetilde{x}_{t,n}\}$ defined by the time instances which make the error $\epsilon_{t,n} = x_t - \widetilde{x}_{t,n}$ equal to zero. Hence, each cluster $j$ is characterized by the random variable $\tau_j \equiv \|t_j - t_{j-1}\|$, with the instances $t_{j-1}$ and $t_j$ referring to subsequent intersection pairs. The random variable $\tau_j$ is named as the *cluster duration*. The empirical distribution of the cluster duration frequencies $P(\tau_j, n)$ can be obtained by ranking the number of clusters $\mathcal{N}(\tau_1, n), \mathcal{N}(\tau_2, n), \ldots, \mathcal{N}(\tau_j, n)$ according to their duration $\tau_1, \tau_2, \ldots, \tau_j$ for each $n$ as:

$$P(\tau_j, n) = \frac{\mathcal{N}(\tau_j, n)}{\mathcal{N}_C(n)}, \tag{1}$$

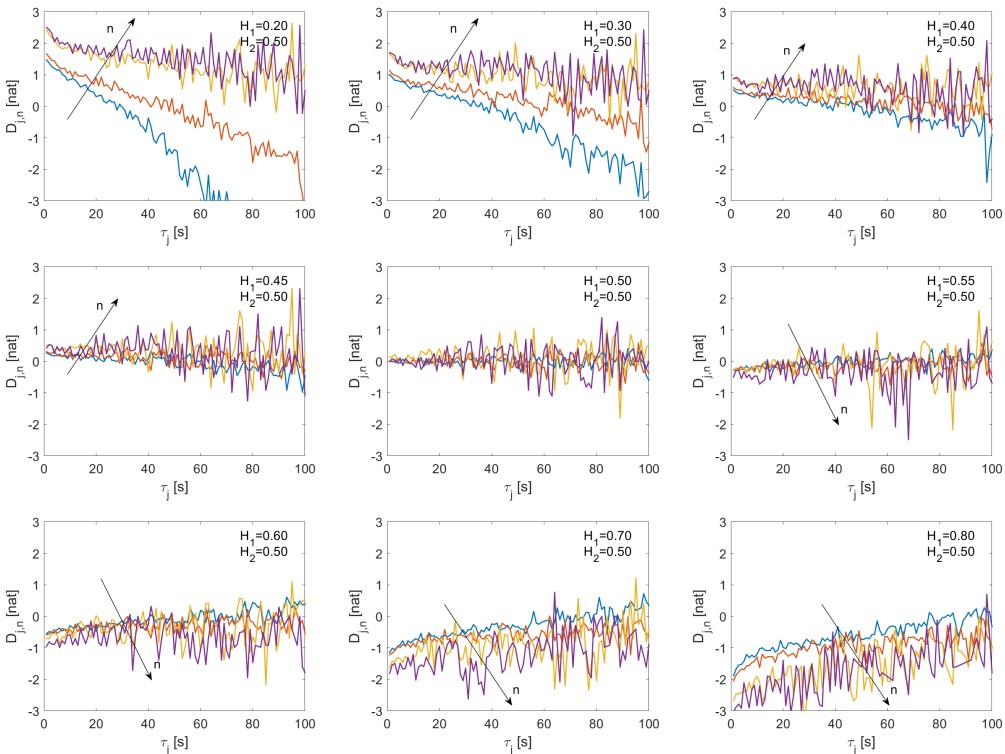

Figure 1: Plot of the quantity $\mathcal{D}_{j,n}$, defined by Eq. (4), as a function of the cluster duration $\tau_j$ for pairs of *fBm* with Hurst exponent $H_1$ and $H_2$. The cluster frequency $P(\tau_j, n)$ is obtained by counting the occurrences of the clusters with duration $\tau_j$ in fractional Brownian motions with Hurst exponent $H_1$. A simple Brownian motion, i.e. a $fBm$ with $H_2 = 0.50$, has been taken to obtain the cluster partition and the model probability $Q(\tau_j, n)$. In the above figures, $H_1$ varies respectively from 0.20 (top-left) to 0.80 (bottom-right). The length of the series is equal to $N = 500000$ for all the graphs. Different curves in each graph refer to different $n$ values ($n = 50, 100, 1000, 2000$) as indicated by the arrow. At large values of the parameter $n$, the curves tend to the asymptotic value $\mathcal{D}_{j,n} = 0$, expected at large $\tau_j$, whereas the curves exhibit a diverging behavior at small values of $\tau_j$. Conversely, at small values of the parameter $n$, the curves tend to the theoretical value expected at small values of $\tau_j$, whereas the curves diverge at large $\tau_j$. The properties of $\mathcal{D}_{j,n}$ are discussed in Section 3 on the basis of the analytical expression derived for continuous random variables.

with $\mathcal{N}_C(n) = \sum_{j=1}^{k(n)} \mathcal{N}(\tau_j, n)$ the number of clusters generated by the partition for each $n$, $k = \sum_{n=1}^{N} \mathcal{N}_C(n)$ the total number of clusters for all the possible values of $n$, and the normalization condition holding as usual:

$$\sum_{n=1}^{N} \sum_{j=1}^{\mathcal{N}_C(n)} P(\tau_j, n) = 1 \,. \tag{2}$$

The *cluster entropy* is defined as:

$$\mathcal{S}_{\mathcal{C}}[P] = -\sum_{j,n} P(\tau_j, n) \log P(\tau_j, n), \tag{3}$$

which is obtained by introducing the cluster frequency $P(\tau_j, n)$ in the Shannon functional.

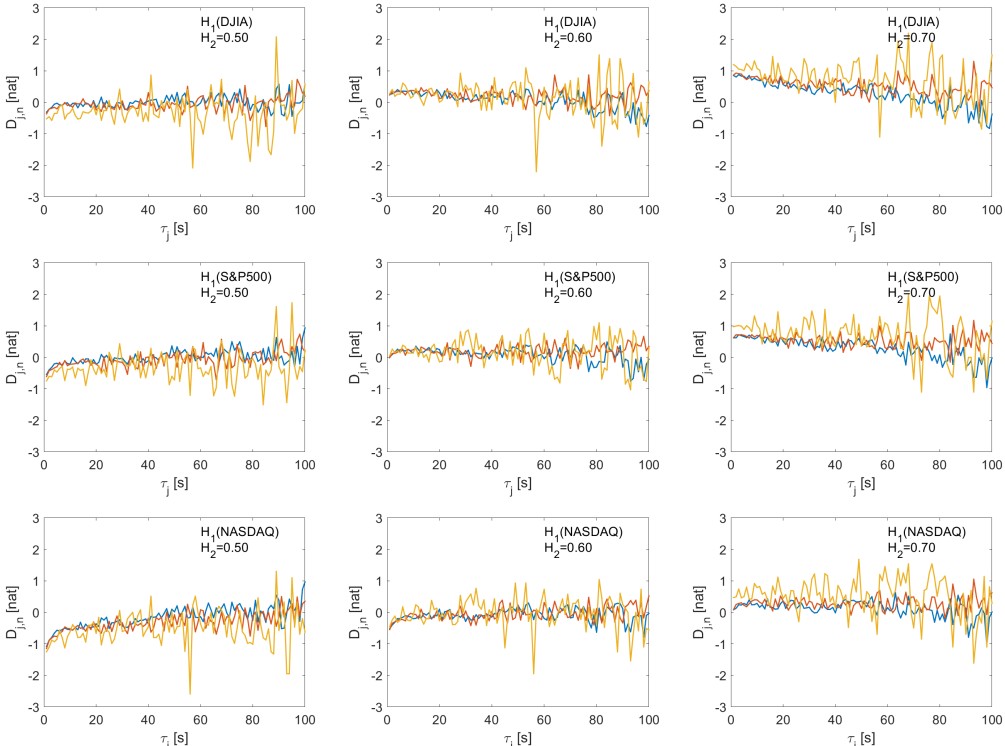

Figure 2: Plot of the quantity $\mathcal{D}_{j,n}$, defined by Eq. (4), vs. cluster duration $\tau_j$. The cluster frequency $P(\tau_j, n)$ has been estimated on the clusters generated by the prices of the DJIA, S&P500, NASDAQ indexes. The model probability $Q(\tau_j, n)$ has been estimated on the clusters generated by a $fBm$ with Hurst exponent $H_2$ ranging from 0.50 to 0.70 and length $N = 492023$ equal to the length of the sampled indexes data. Different curves in each graph refer to different values of the parameter $n$ (respectively $n = 50$ blue; $n = 100$ orange; $n = 1000$ yellow).

In this work, the *relative cluster entropy* or *cluster divergence* $\mathcal{D}_{\mathcal{C}}[P\|Q]$ is proposed to quantify the wrong information yield when a model probability distribution $Q$ is assumed in place of the empirical probability distribution $P$. A measure of distinguishability between two probability distributions $P$ and $Q$ is the *Kullback-Leibler divergence*, defined for discrete variables as $\mathcal{D}_{\mathcal{KL}}[P\|Q] = \sum_j P_j \log\left(P_j/Q_j\right)$, with the conditions $\operatorname{supp}(P) \subseteq \operatorname{supp}(Q)$ and $\mathcal{D}_{\mathcal{KL}}[P\|Q] \geq 0$, with $\mathcal{D}_{\mathcal{KL}}[P\|Q] = 0$ for $P = Q$. Then, the *minimum relative entropy* principle can be adopted as optimization criterion for model selection and statistical inference.

The quantity $\mathcal{D}_{j,n}[P\|Q]$ is defined for each macrostate in terms of the cluster durations $\tau_j$ as follows:

$$\mathcal{D}_{j,n}[P\|Q] = P(\tau_j, n) \log \frac{P(\tau_j, n)}{Q(\tau_j, n)}, \tag{4}$$

where the index $j$ refers to the set of clusters with duration $\tau_j$ generated by the partition for a given $n$. The cluster frequencies $P(\tau_j, n)$ and $Q(\tau_j, n)$ satisfy the condition $\operatorname{supp}(P) \subseteq \operatorname{supp}(Q)$. By using Eq. (4) and summing $\mathcal{D}_{j,n}[P\|Q]$ over all the accessible cell states , the *relative cluster entropy* is written as:

$$\mathcal{D}_{\mathcal{C}}[P\|Q] = \sum_{n=1}^{N} \sum_{j=1}^{\mathcal{N}_C(n)} P(\tau_j, n) \log \frac{P(\tau_j, n)}{Q(\tau_j, n)}, \tag{5}$$

where the index $j$ runs over the clusters obtained by each partition with size $n$, which in turn runs over the allowed set of time window values, $n \in (1, N)$.

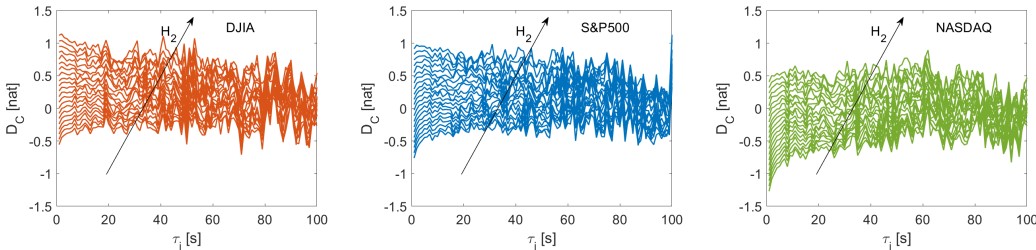

Figure 3: Plot of the quantity $\mathcal{D}_{\mathcal{C}}[P||Q]$, defined by Eq. (5), vs. cluster duration $\tau_j$. The curves are obtained by summing the quantities $\mathcal{D}_{j,n}[P||Q]$, as those shown in Fig. 2, over the parameter $n$ for the prices of DJIA, S&P500, NASDAQ. Each curve in the figures corresponds to the cluster divergence with the probability $P(\tau_j, n)$ referred to the market price series $p_t$ and the model probability $Q(\tau_j, n)$ referred to fBms with Hurst exponent $H_2$ ranging from 0.50 to 0.70 with step 0.1 as indicated by the arrow.

To exemplify how the relative cluster entropy could be applied in practice, pairs of artificially generated fractional Brownian motions (fBms) are analysed in terms of the *relative cluster entropy* defined by Eqs. (4-5). Fractional Brownian motions (fBms) $x_t^H$ with $t \geqslant 0$ are power-law correlated stochastic processes, defined by a centered Gaussian process with stationary increments and covariance given by $<x_s^H x_t^H> = \frac{1}{2}\left(t^{2H} + s^{2H} - |t-s|^{2H}\right)$ with $H \in (0,1)$ the Hurst exponent. Power-law behaviour of the correlation function implies slow memory decay and non-Markovianity. Synthetic fBm sequences have been generated with assigned Hurst exponent $H$ and length $N$ by using the FRACLAB code [22]. The cluster frequencies $P(\tau_j, n)$ and $Q(\tau_j, n)$ have been estimated by counting the number of clusters with duration $\tau_j$ and window $n$ for each $fBm$.

Fig. 1 shows a few examples of plots of the quantity $\mathcal{D}_{j,n}$, defined by Eq. (4). $\mathcal{D}_{j,n}$ is estimated for cluster frequency $P$, obtained from clusters generated in $fBms$ with $H_1$ varying from 0.20 (top-left) to 0.80 (bottom-right) with step 0.05, and model distribution $Q$ obtained from uncorrelated Brownian paths, i.e. $fBms$ with $H_2 = 0.50$. The values of the Hurst exponents correspond respectively to correlation exponents $\alpha_1 = 2 - H_1$ ranging from 1.80 to 1.20, whereas $\alpha_2 = 2 - H_2$ is kept constant and equal to 1.50. The quantity $\mathcal{D}_{j,n}$ shows characteristic deviations with respect to the null hypothesis corresponding to a fully random process with $H_2 = 0.5$. In particular, at small values of the cluster duration $\tau_j$, the quantity $\mathcal{D}_{j,n}$ takes positive and negative values respectively for fBms with $0.5 < H_1 < 1$ and $0 < H_1 < 0.5$. As the cluster duration $\tau_j$ increases, $\mathcal{D}_{j,n}$ tends to reach the horizontal axis implying that the divergence between the distributions become negligible for very large clusters. Note in particular the three panels of the middle row in Fig. 1 showing the results obtained for fractional Brownian motions with $H_1 = 0.45$, $H_1 = 0.50$ and $H_1 = 0.55$ with respect to the simple Brownian path, i.e. the $fBm$ with $H_2 = 0.50$, taken as the model. Thus, fBm pairs with close values of $H_1$ and $H_2$ correspond to more realistic experimental conditions. Inference problems with data sequences featuring correlation exponents statistically close to each other and small deviations from the model distribution should be reasonably expected in the cases of practical interest.

To further illustrate how the proposed method operates with real-world data, price series $\{p_t\}$ of Dow Jones Industrial Average (DJIA), Standard and Poor 500 (S&P500), National Association of Securities Dealers Automated Quotations Composite (NASDAQ), are considered. Data include tick-by-tick prices from January to December 2018. Details (Ticker; Extended name; Country; Currency; Members; Length) provided by Bloomberg [23]. Raw data prices $\{p_t\}$ have different lengths ($N_{DIJA} = 5749145$, $N_{S\&P500} = 6142443$, $N_{NASDAQ} = 6982017$). To perform the relative cluster entropy analysis over comparable data sets, raw data prices are

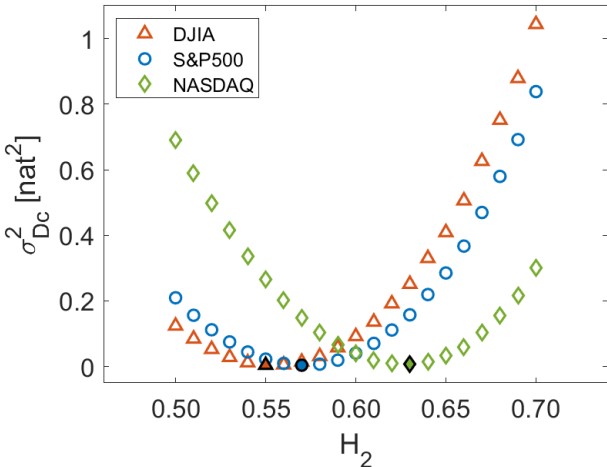

Figure 4: Plot of the quantity $\sigma^2_{\mathcal{D}_C}$ defined by Eq. (6) for the relative cluster entropy curves plotted in Fig. 3 vs. the Hurst exponent $H_2$ of the model distribution $Q(\tau_j, n)$. The quantity $\sigma^2_{\mathcal{D}_C}$ is the variance of $\mathcal{D}_C$ with respect to 0 (the null hypothesis for $P = Q$) over the cluster lifetime interval $1 < \tau_j < 20$. Each point is evaluated by using the definition given in Eq. (6) for each market and for each $fBm$ with assigned Hurst exponent $H_2$. The Hurst exponent $H_2$ of the model distribution $Q(\tau_j, n)$ ranges between 0.50 and 0.70 with step 0.01. The minimum value of the variance is obtained for $H_1 = 0.55$ (DJIA), $H_1 = 0.57$ (S&P500) and $H_1 = 0.63$ (NASDAQ).

sampled to yield equally spaced data sequences with equal length $N$. The cluster frequency $P(\tau_j, n)$ is estimated by counting the clusters generated in the market price series. $Q(\tau_j, n)$ is estimated by counting the clusters generated in synthetic stochastic processes assumed as a model. In this analysis, the divergence between each price series, with unknown correlation exponent, and artificially generated samples of fractional Brownian motions *fBms* with assigned Hurst exponent $H_2$, is considered. Results of the analysis are plotted in Fig. 2, showing the relative cluster entropy for the three markets. Several samples of the divergence obtained for different values of the parameter $n$, shown in Fig. 2, have been summed over the parameter $n$, with same interval of cluster duration $\tau_j$. Fig. 3 shows the relative cluster entropy $\mathcal{D}_C[P||Q]$ for the data shown in Fig. 2.

To infer the optimal probability distribution $P$, the *minimum relative entropy* principle is implemented non-parametrically on the values plotted in Fig. 3. To this purpose, the variance $\sigma^2_{\mathcal{D}_C}$ of $\mathcal{D}_C[P||Q]$ around the value $\mathcal{D}_C[Q||Q]$ (the null hypothesis for $P = Q$) is written as follows:

$$\sigma^2_{\mathcal{D}_C} \equiv \frac{1}{k-1} \sum_{j=1}^{k} \left[ \mathcal{D}_C[P||Q] - \mathcal{D}_C[Q||Q] \right]^2 , \tag{6}$$

where the sum runs over the total number of clusters obtained by the partition process. By using the value $\mathcal{D}_C[Q||Q] = 0$, Eq. (6) writes:

$$\sigma^2_{\mathcal{D}_C} = \frac{1}{k-1} \sum_{j=1}^{k} \left[ \mathcal{D}_C[P||Q] \right]^2 . \tag{7}$$

The quantity $\sigma^2_{\mathcal{D}_C}$ corresponds to the mean square value of the area of the region between the curve $\mathcal{D}_C[P||Q]$ and the horizontal axis ($\mathcal{D}_C[Q||Q] = 0$). Given the linearity of the relative cluster entropy operator, $\sigma^2_{\mathcal{D}_C}$ exhibits a quadratic behaviour with the typical asymmetry of the Kullback-Leibler entropy. The quadratic functional can be easily used to estimate the

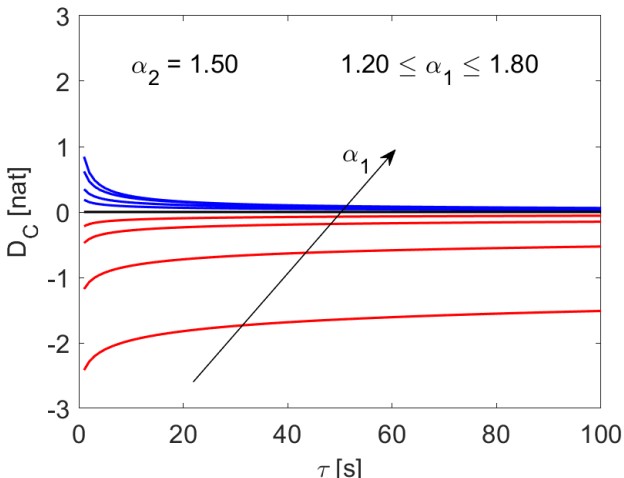

Figure 5: Plot of the quantity $\mathcal{D}_{\mathcal{C}}[P\|Q]$, defined by Eq. (11), vs. the cluster duration $\tau$. Blue curves correspond to a power law probability distribution $P(\tau)$ with $\alpha_1$ ranging between $1.55 \div 1.80$. The model probability distribution $Q(\tau)$ is a power law with correlation exponent $\alpha_2 = 1.50$, the same for all the curves plotted here. Red curves correspond to a power law probability distribution with $\alpha_1$ ranging between $1.20 \div 1.45$. The black line corresponds to the null hypothesis $\mathcal{D}_{\mathcal{C}}[P\|P] = 0$ obtained with $\alpha_1 = 1.50$ and $\alpha_2 = 1.50$.

minimum. The minimization criterion provided by Eq. (6) has been applied to the data shown in Fig. 3 to yield the best estimate of the correlation degree of the market prices. The value of the Hurst exponent for the series of the prices $\{p_t\}$ has been deduced from the value of $H_2$ for which $\sigma_{\mathcal{D}_C}^2$ takes its minimum, implying $H_1 = H_2$. By using this rule, $H_1 = H_2 = 0.55$, $H_1 = H_2 = 0.57$, and $H_1 = H_2 = 0.63$ have been found respectively for DJIA, S&P500 and NASDAQ. The minimization outcomes are plotted in Fig. 4 for the markets shown in Fig. 3.

## 3 Discussion and Conclusion

In this Section, the *relative cluster entropy* is extended to continuous random variables. For $N_{\mathcal{C}}(n) \to \infty$, the characteristic size of generated clusters $\mathcal{C}$ behaves as continuous random variables $\tau \in [1, \infty]$ with probability distribution function $P(\tau)$ varying as a power-law [17, 18]. By taking the limits $P(\tau_j) \to P(\tau)d\tau$ and $Q(\tau_j) \to Q(\tau)d\tau$, Eq. (5) can be written for continuous random variables in the form of an integral:

$$D_C[P(\tau)\|Q(\tau)] = \int P(\tau)\log\frac{P(\tau)}{Q(\tau)}d\tau, \tag{8}$$

with $\tau \in [1, \infty]$. We are interested in the situations where the probability distributions are power-law functions, i.e. for $P(\tau)$ and $Q(\tau)$ respectively in the form:

$$P(\tau) = (\alpha_1 - 1)\tau^{-\alpha_1} \qquad Q(\tau) = (\alpha_2 - 1)\tau^{-\alpha_2}, \tag{9}$$

where $\alpha_1$ and $\alpha_2$ are the correlation exponents, $\alpha_1 - 1$ and $\alpha_2 - 1$ are the normalization constants for $\tau \in [1, \infty]$. By using Eqs. (9), Eq. (8) writes:

$$D_C[P(\tau)\|Q(\tau)] = \int (\alpha_1 - 1)\tau^{-\alpha_1} \log \frac{(\alpha_1 - 1)\tau^{-\alpha_1}}{(\alpha_2 - 1)\tau^{-\alpha_2}}d\tau, \tag{10}$$

that after integration becomes:

$$D_C[P(\tau)||Q(\tau)] = \tau^{1-\alpha_1} \left( \log \frac{\alpha_1 - 1}{\alpha_2 - 1} + \left( \log \tau^{(\alpha_1 - \alpha_2)} + \frac{\alpha_1 - \alpha_2}{1 - \alpha_1} \right) \right) + \text{constant}, \qquad (11)$$

where the integration constant is equal to zero by setting $D_C[P||P] = 0$. By estimating the definite integral over the interval $[1, \infty]$, one obtains:

$$D_C[P||Q] = \log \frac{\alpha_1 - 1}{\alpha_2 - 1} + \frac{\alpha_1 - \alpha_2}{1 - \alpha_1}, \qquad (12)$$

that for $\alpha_1 = \alpha_2$, i.e. for the distribution $P$ coincident with the model distribution $Q$, provides $D_C[P||Q] = 0$.

$D_C[P||Q]$ quantifies the divergence between $P(\tau)$ and $Q(\tau)$, respectively true and model distribution, as a function of the cluster lifetime $\tau$ in terms of the pair of correlation exponents $\alpha_1$ and $\alpha_2$. Eq. (11) is plotted as a function of $\tau$ for different values of the exponents $\alpha_1$ and $\alpha_2$ in Fig. 5. At small values of the cluster duration ($\tau \to 1$), $D_C[P||Q]$ is strongly dependent on the difference of the power-law exponent $\alpha_1$ with respect to the exponent $\alpha_2$ of the model distribution. Conversely, as the cluster duration increases ($\tau >> 1$), $D_C[P||Q]$ becomes negligible. The decay can be understood by considering that as $\tau$ increases the cluster becomes disordered as a consequence of the spread of the distribution and the onset of finite-size effect. The correlation vanishes as the process becomes almost fully uncorrelated. The behaviour of the cluster distribution divergence obtained by using continuous variables is consistent with the empirical tests performed on discrete data sets. In particular, the behaviour shown by the fractional Brownian motions with different correlation exponents discussed in the Section II is reproduced by the curves shown in Fig. 5, ensuring that the approach is sound and robust.

The *relative cluster entropy* can be therefore exploited to estimate the deviation of the power law exponent corresponding respectively to experimental and model probability distributions.

Long-range correlated processes obeying power-law distributions occur frequently in complex system data related to several natural and man-made phenomena. Due to their ubiquity, the extent of long-range correlation and the scaling exponents are relevant to many disciplines, though several difficulties are met for their estimation which require suitable computational procedures to be carefully implemented [21]. A random variable $x$ obeys a power law if it is drawn from a probability distribution $p(x) \propto x^{-\alpha}$ with $\alpha > 1$ the correlation exponent. Empirical real-world data barely follow a power-law for all the values of $x$. Due to normalization requirements and finite-size effects, ideal power-law behaviour usually holds at values greater than some minimum $x_{\min}$ up to a maximum $x_{\max}$. An exponential cut-off is often artificially introduced to account for the deviation from the ideal power-law behaviour $x^{-\alpha}e^{-\lambda x}$.

The non-parametric minimization of the relative entropy has some advantages compared to the parametric approaches, whose implementation requires normality of the random variables and knowledge of the first two moments of the distribution for the calculation of the Lagrange multipliers. The proposed *relative cluster entropy* approach yields the optimal value of the correlation exponent $\alpha$ without relying on the estimate of the slope in a log-log plot and thus is robust against computational biases which usually affect least-squares estimates.

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
