# Peer review of "Relative cluster entropy for power-law correlated sequences"

_SciPost Physics, doi:SciPost Phys. 13, 076 (2022)_

## Round 1 · Referee Report · Anonymous (Referee 1) · 2022-7-7

Strengths

Gives a robust method for quantifying non-Markovian (memory) effects in important practical times series.

Weaknesses

The motivation for some of the information theoretic tools used is not well explained.

Report

This paper presents an interesting new information theoretic technique for diagnosing memory effects in practical time series. It generally meets the acceptance criteria for the journal however more explanatory material would help a lot in allowing readers to decide on the applicability of the method to their own work.

Requested changes

  1. Give a clear detailed justification for the clustering method used for applying the relative entropy functional. At present the method appears rather ad hoc. Is the method some kind of coarse graining of the discrete time time series?

  2. Justify more clearly the use of equation (6) in the manuscript. Intuitively the relative entropy between coarse grained versions of the target time series and the model fractional Brownian motion time series would be the obvious way to proceed. Equation (6) appears superficially at least to be more complicated than this straightforward approach. Why?

  • validity: good
  • significance: high
  • originality: high
  • clarity: ok
  • formatting: excellent
  • grammar: excellent

Author:  Anna Carbone  on 2022-07-29  [id 2696]

(in reply to Report 1 on 2022-07-07)
Category:
answer to question

Thank you very much for the fruitful comments. Part of the text below clarifying these comments will be included in the revised manuscript. Here below our response to the questions 1 and 2 follows:

Answer to Question 1:

The cluster entropy $\mathcal{S_{C}}(P)$ and the relative cluster entropy $\mathcal{D_{C}}(P \| Q) $ can be interpreted as an information measure over a partition generated by a coarse-grained mapping of the two-dimensional phase-space spanned by a particle (e.g. a simple Brownian path $\{x_t \}$). According to Gibbs’ original idea at the core of the information entropy concept, a coarse grained description is defined by smoothing out the fine details of the system hence increasing the observer’s ignorance about the exact microstate of the system. As the structure of the system becomes blurry, randomness and entropy increase. Coarse-grained descriptions are obtained by performing a local average over each cell of the partition in phase space: average temperature or free energy are classical examples.

In the information clustering approach proposed in our manuscript, the coarse grained description of the particle path $\{x_t \}$ is obtained by a local average $\widetilde{x}_{t,n}$ over the phase-space cells where $n$ indicates the cell size. The linear regression $ {x_{t}}= \widetilde{x}_{t,n}+\epsilon_{t,n} $ yields the error in the estimate $\epsilon_{t,n} ={x_{t}}- \widetilde{x}_{t,n}$, which quantifies the loss of structure and information of the coarse grained description. The values of $t$ when $\epsilon_{t,n}=0$ corresponds to no loss of information and to the minimum value of the cluster entropy $\mathcal{S_{C}}$ . The errors $\epsilon_{t,n}$ generate a random sequence of macrostates, which ultimately define for each $n$ a finite partition $\mathcal{C}_{n,j}=\left\{C_{n, 1}, C_{n, 2}, \ldots\right\}$ of macrostates of the particle path clustered between the values of $t$ when $\epsilon_{t,n}=0$.
$\mathcal{S_{C}}(P)$ is bounded, involves integrating over cell components, ranges from a minimum value ($0$) at the finest-grained description (corresponding to smallest clusters) towards the maximum value ($log N$) at the coarsest-grained description. In other words, $\mathcal{S_{C}}(P)$ provides a tool to estimate the entropy or randomness increase as the partition becomes coarser and coarser.
$\mathcal{D_{C}}(P \| Q) $ is also bounded, involves integration over cells, ranges between the maximum value given by Eq.(11) and the minimum value which is zero for $P=Q$.

Answer to Question 2:

The notation in Eq. (6) of the submitted manuscript was really unfortunate as we used the symbol $D_V $ to indicate the variance of the relative cluster entropy with respect to the null hypothesis of zero divergence. In this answer we provide the revised notation for Eq.(6) where the variance of $\mathcal{D}_{C}[P|| Q ] $ is indicated by $\sigma^2_{\mathcal{D}_C} $ (thus using the common notation) . We also argue on the reason why the quadratic behaviour of the variance conveniently provides, in a straightforward and accurate way, the minimum value of the relative cluster entropy (and ultimately the corresponding optimal power law exponent of the analysed data) rather than using directly the value of $\mathcal{D}_{C}[P|| Q ] $. For the sake of completeness, we report below the updated text and notation:

"To obtain the best estimate of the probability distribution $P$, the minimum relative entropy principle is implemented non-parametrically on the values plotted in Fig.4 of the submitted manuscript. To this purpose, the variance $\sigma^2_{\mathcal{D}_C} $ of $\mathcal{D}_{C}[P|| Q ] $ around the value $\mathcal{D}_{C}[Q|| Q] =0 $, i.e. the null hypothesis expected for $P=Q$ is estimated:
\begin{equation*}
{\sigma^2_{\mathcal{D}_C}} \equiv \frac{1}{k-1} \sum_{j=1}^k
\left[\mathcal{D}_{C}[P|| Q]- \mathcal{D}_{C}[Q||Q]\right]^2 \equiv \frac{1}{k-1} \sum_{j=1}^k
\left[\mathcal{D}_{C}[P|| Q]\right]^2
\end{equation*}
where the sum runs over the range of cluster indexes obtained by the partition process. The right side of Eq.(6) corresponds to the mean square value of the area of the region between the curve $\mathcal{D}_{C}[P||Q]$ and the horizontal axis. Given the linearity of the relative entropy operator, $\sigma^2_{\mathcal{D}_C} $ is positive defined has a quadratic behaviour characterized by the typical asymmetry of the Kullback-Leibler entropy and thus can be easily used to estimate the minimum of the relative cluster entropy."

---

## Round 1 · Referee Report · Anonymous (Referee 2) · 2022-7-27

Strengths

The manuscript presents an innovative approach to the estimate of the information loss between two distributions based on the Kullback-Leibler distance

Weaknesses

The presentation in some points can be improved

Report

The ability of the method mainly derives from the innovative partition of the phase space performed in 2d where the two dimensions are the cluster duration and rolling window. The results are justified from the computational point of view with the analysis of synthetic and real world data (Section II) and from the analytic point of view by discussing the findings in terms of continuous variable distribution (Section III). The results are interesting and important from both the fundamental and applied perspectives. The approach could herald several further applications beyond the fractional Brownian motions and financial time series considered in this work.

Thus I strongly recommend publication.

Nevertheless some sentences are too long or redundant and some minor defects (typos or graphical quality) could be removed. Hence, I would require a few minor corrections that should be considered before publication mainly to improve the quality of the presentation .

Requested changes

  1. The normalization condition (Eq.(3)) is referred to the probability distribution function Eq. (1), hence it should be reported just after Eq. (1), before Eq.(2).

  2. The arrows in the nine panels of Fig. 3 do not help at all the reader to catch the values of the parameter $n$ as the three curves overlap. I would suggest removing the arrows, while just adding a text linking each color to the values of $n$ in the caption.

  3. Eqs. (7-10) refer to the relative entropy for the continuous variable $\tau$ while Eqs. (4-5) refer to the relative entropy for the discrete variable $\tau_j$, in order to avoid confusion between the two sets of equation please just use time roman $D$ in Eqs. (7-10) instead of calligraphic D (already used in Eqs. (4-5)).

  4. I believe that three different panels in Fig. 5 are not needed. I would recommend it possible to present together the left, middle and right panel in one single graph.

  • validity: top
  • significance: top
  • originality: top
  • clarity: good
  • formatting: good
  • grammar: excellent

Author:  Anna Carbone  on 2022-08-01  [id 2701]

(in reply to Report 2 on 2022-07-27)
Category:
answer to question

Thanks a lot for reading the manuscript and acknowledging the importance of the approach from both the fundamental and applied perspectives.

All the four minor corrections requested in your report will be implented in the revised version of the manuscript.

---

## Round 1 · Referee Report · Anonymous (Referee 1) · 2022-7-29

Strengths

Interesting new technique for measuring the non-Markov nature of practical stochastic processes.

Weaknesses

None

Report

Concerns in earlier report have been adequately addressed.

Requested changes

None

  • validity: -
  • significance: -
  • originality: -
  • clarity: -
  • formatting: -
  • grammar: -

Author:  Anna Carbone  on 2022-08-01  [id 2702]

(in reply to Report 3 on 2022-07-29)
Category:
answer to question

Thank you very much for the report and for considering the proposed approach interesting.
We deeply appreciate the time spent to read our work.

---

## Round 1 · Referee Report · Anonymous (Referee 3) · 2022-8-7

Report

The manuscript presents a new measure, the relative cluster entropy, to analyze long-range correlated datasets. The authors demonstrate the measure using synthetic data (fractional Brownian motions) and empirical data (US market time series). The topic is interesting and the results are sound, therefore, I recommend the acceptance of this manuscript, after the authors take into account the requested change.

Requested changes

I believe the authors should improve the motivation of their work. In the introduction the authors explain previous measures (cluster entropy and differential cluster entropy) but do not explain which limitations or drawbacks these measures have, that the authors want to address by proposing the new measure (relative cluster entropy). In other words, why this new measure is needed? which are the problems that this new measure is aimed at?

  • validity: good
  • significance: good
  • originality: good
  • clarity: good
  • formatting: -
  • grammar: -

Author:  Anna Carbone  on 2022-08-08  [id 2716]

(in reply to Report 4 on 2022-08-07)

Thank you very much for considering the topic interesting, the results sound, and for recommending the acceptance of our manuscript. We deeply appreciate the time you spent to read our work and the requested changes of better clarifying the motivation of our study, whose inclusion in the Introduction of the revised version will certainly add value to the manuscript.

The text below summarizes the motivations for proposing the "relative cluster entropy" $\mathcal{D_{C}}(P \| Q) $ and the main differences with respect to the simple "cluster entropy" $\mathcal{S_C} (P)$.
This text will be included in the Introduction of the revised manuscript.

Answer to question:

$\mathcal{S_C} (P)$ holds the main properties of the Shannon functional, in particular, $ 0 \leq \mathcal{S_C} (P) \leq \ln N $. If $P$ is a distribution concentrated on a single cluster value, $\mathcal{S_C} (P) = 0 $ corresponds to the minimum uncertainty on the outcome of the cluster, the random variable of interest. If $P$ is a fully developed power-law distribution, $\mathcal{S_C} (P)=\ln N$ corresponds to the maximum uncertainty obtained as the power-law distribution spreads over a broad range of cluster values. Thus, according to the Shannon interpretation, $\mathcal{S_C} (P)$ is a measure of randomness of all the possible cluster outcomes or equivalently a measure of the variability associated with the random variable.

In this work, we go beyond the simple "measure of uncertainty of the random variable (the cluster size)" provided by $\mathcal{S_C} (P)$.

We develop an "inference method for hypothesis testing of a general class of models" underlying the stochastic phenomena under investigation. We put forward the $\mathcal{D_{C}}(P \| Q) $ cluster divergence with the first argument $P$ the empirical distribution and the second argument $Q$ a model within a broad class of probability distributions. It is a metric on the space of probability distribution, called a divergence rather than a distance since it does not obey symmetry and triangle inequality.

The asymmetry of the relative entropy reflects the asymmetry between data and models, hence it can be used for inference purposes on the model underlying a given distribution. If $\mathcal{D_{C}}(P \| Q) >> 0 $, the hypothesis likelihood is very low and, unless the quality of the empirical data should be questioned, the model distribution $Q$ must be rejected. The higher $\mathcal{D_{C}}(P \| Q) $, the lower the likelihood of the hypothesis. This property allows to use $\mathcal{D_{C}}(P \| Q) $ for hypothesis test about a model distribution $Q$. If the hypothesis on the model were true, $P$ should fluctuate around its expected value $Q$, with fluctuations of limited amplitude, with occurrence probability greater than the significance level, permitting the acceptance of the hypothesis on the model $Q$.

The ability of the divergence to select an optimal distribution might be relevant in several contexts and in particular complex phenomena obeying power-law distributions. The accuracy of the estimation of the power law exponent is still actively debated by the scientific community (Ref. Aaron Clauset et al, SIAM review 2009).

---

## Round 2 · List of Changes

Change 1: A short text clarifying the meaning of the approach and its relation with the coarse-grained is now added in the Introduction.
Change 2: Notations in Eq. (6) have been improved. For the sake of clarity, Eq. (6) has been split over two lines.
Change 3: Eq. (3) has been moved after Eq. (1) and before Eq. (2). Eq. (3) is now labelled Eq. (2).
Change 4: The arrows in the nine panels of Fig. 3 have been removed. A text linking each colour to the parameter n is now added in the caption.
Change 5: The left hand of equations in Section III are now in “math roman font” instead of "mathcal".
Change 6: The three panels in Fig. 5 have been merged together in one single panel.
Change 7: A short text, clarifying the main motivations of the work, is now included in the Introduction.

---

## Editorial Decision

published